# In Silico Screening of Bacteriocin Gene Clusters within a Set of Marine *Bacillota* Genomes

**DOI:** 10.3390/ijms25052566

**Published:** 2024-02-22

**Authors:** Rabeb Teber, Shuichi Asakawa

**Affiliations:** 1Laboratory of Aquatic Molecular Biology and Biotechnology, Department of Aquatic Bioscience, Graduate School of Agricultural and Life Sciences, The University of Tokyo, Bunkyo, Tokyo 113-8657, Japan; rabeb.teber@gmail.com; 2Signal Peptidome Research Laboratory, Department of Aquatic Bioscience, Graduate School of Agricultural and Life Sciences, The University of Tokyo, Bunkyo, Tokyo 113-8657, Japan

**Keywords:** bacteriocin, in silico, BAGEL4, marine *Bacillota* (*Firmicutes*)

## Abstract

Due to their potential application as an alternative to antibiotics, bacteriocins, which are ribosomally synthesized antimicrobial peptides produced by bacteria, have received much attention in recent years. To identify bacteriocins within marine bacteria, most of the studies employed a culture-based method, which is more time-consuming than the in silico approach. For that, the aim of this study was to identify potential bacteriocin gene clusters and their potential producers in 51 marine *Bacillota* (formerly *Firmicutes*) genomes, using BAGEL4, a bacteriocin genome mining tool. As a result, we found out that a majority of selected *Bacillota* (60.78%) are potential bacteriocin producers, and we identified 77 bacteriocin gene clusters, most of which belong to class I bacteriocins known as RiPPs (ribosomally synthesized and post-translationally modified peptides). The identified putative bacteriocin gene clusters are an attractive target for further in vitro research, such as the production of bacteriocins using a heterologous expression system.

## 1. Introduction

The World Health Organization (WHO) has considered the rise of antibiotic resistance as one of the major threats to human and animal health because its emergence has jeopardized the treatment of common infectious diseases [1]. Due to their potential as alternatives to antibiotics, bacteriocins have attracted a lot of attention. Bacteriocins are ribosomally synthesized antimicrobial peptides produced by bacteria with either narrow- or broad-spectrum activity. They are generally categorized into three classes: class I encompasses post-translationally modified peptides known as RiPPs (ribosomally synthesized and post-translationally modified peptides); class II, unmodified and heat-stable peptides (<10 kDa); and class III, heat-labile proteins (>30 kDa) [2].

To identify bacteriocins within bacteria, a culture-based method was originally used. Compared to the culture-based method, the in silico approach is known to be less time-consuming and costly, which has led to its widespread use in the past decade [3]. This increased interest in the in silico approach is also due to the availability of a large amount of metagenomics data [4] and of bacteriocin mining tools such as BAGEL4. BAGEL4 predicts bacteriocins gene clusters (GCs) and their producers by examining the bacterial genome for the genes encoding the precursor peptide and/or for the associated context genes involved in the bacteriocin machinery biosynthesis, which are usually located in proximity to the precursor peptide [5,6]. These associated context genes are related to the post-translational modification of the precursor peptide in the case of RiPPs, to the transport and release of the mature bacteriocin, and to the self-immunity, protecting the producer from its own bacteriocins [6]. Based on the area of interest identified, BAGEL4 categorizes the predicted bacteriocins within one of the three classes. Therefore, BAGEL4 can identify the putative bacteriocins GCs through the core-peptide database or by mining the associated context genes through HMM motifs [5].

To date, the in silico prediction of bacteriocins gene clusters has been performed within bacteria found in the human gastrointestinal tract [4], rumen [7,8] and artisanal cheese [9], as well as within the bacterial genus *Geobacillus* [3]. Although both terrestrial and aquatic bacteriocins could present a promising alternative to antibiotics [10], very few studies related to the exploration of marine bacteriocins have been carried out [11,12]. A bacteriocin, BaCf3, produced by the marine strain *Bacillus amyloliquefaciens* was found to be thermostable and active against an antibiotic-resistant strain, *Bacillus circulans*, through an in vitro study [13]. Another recent study that combined in vitro and in silico approaches to investigate the bacteriocinogenic potential of bacterial strains isolated from a deep-sea fish showed that the strains with antibacterial activity have a wide variety of GCs, with the RiPP being the most commonly identified [11]. The marine habit is known to be an untapped reservoir of new and unique natural products, including antimicrobial peptides. This is attributed to the uniqueness of the organisms producing them and to the extremophilic biodiversity in the marine environment [12,14]. Among the bacterial phyla widely distributed in the marine ecosystem, *Bacillota* (formerly named as *Firmicutes*) is recognized for its potential antibacterial activity [14,15]. However, out of the microbial natural products from both marine and terrestrial *Bacillota*, only 9% were uniquely associated with marine *Bacillota* (*Firmicutes*) [14]. To the best of our knowledge, no studies have determined bacteriocin gene clusters in marine *Bacillota* (*Firmicutes*) through the in silico method. In that respect, the aim of the present study was to screen the 51 complete *Bacillota* genomes, from different marine sources, for potential bacteriocin-encoding genes, using BAGEL4, and to assess the abundance and phylogenetic distribution of different identified bacteriocins GCs within a set of marine *Bacillota* species.

## 2. Results

### 2.1. Distribution of Putative Bacteriocin Gene Clusters in the Selected Marine Bacillota Genomes

The in silico analysis of 51 *Bacillota* genomic sequences using BAGEL4 revealed that 31 strains (60.78%) possess at least one putative bacteriocin gene cluster, whereas no putative bacteriocin gene was identified in 39.22% of the strains selected, suggesting the prevalence of putative bacteriocin GCs within the chosen marine *Bacillota* genomes. The preliminary BAGEL4 screening revealed the identification of 82 bacteriocins GCs. However, among the predicted GCs, five putative bacteriocin GCs did not harbor any precursor-encoding gene or essential surrounding biosynthetic-associated genes. The absence of those genes was confirmed upon further investigation through BlastP to the NCBI non-redundant protein database (nr database). Consequently, one sactipeptide GC (*Paenibacillus durus* DSM 1735), three lasso peptide GCs (*Paenibacillus durus* DSM 1735, *Paenibacillus crassostreae* LPB0068 and *Petrocella atlantisensis* 70B-A), and one class II bacteriocin (UviB) GC (*Paenibacillus* sp. CAA11) were excluded. In total, the 31 potential *Bacillota* genomes were found to encode 77 bacteriocins GCs in total, varying from 1 to 8 gene clusters per strain and belonging to different bacteriocins classes, namely nine subclasses of RiPPs, class II bacteriocin and class III bacteriocin. The phylogenetic distribution of putative bacteriocin GCs in *Bacillota* strains is represented in Figure 1. Excluding the three *Streptococcus iniae* strains, all the other selected potential *Bacillota* genomes were shown to harbor different numbers and types of bacteriocin GCs (Figure 1). *Bacillus* spp., the most representative species in our study and one of the most important *Bacillota* species [15], represents, in our study, an example of the variation in bacteriocin classes at the intra-species and especially at the inter-species level (Figure 1).

### 2.2. Abundance of Different Classes of Bacteriocins Gene Clusters Identified by In Silico Analysis

Among the 77 putative bacteriocin GCs identified by BAGEL4, 60 class I bacteriocin GCs (77.92%) were predicted, indicating that the RiPPs are the most prevalent putative peptide in our study, followed by 16 class II bacteriocin GCs and 1 class III GC. 

#### 2.2.1. Class I: The RiPPs (Ribosomally Synthesized and Post-Translationally Modified Peptides)

The RiPPs have a common biosynthetic GC, and they are divided into diverse subclasses based on their post-translational modification sites [16]. The GC encodes the RiPPs synthesis pathway that started with the ribosomal synthesis of a precursor peptide, which is generally composed of an *N*-terminal leader peptide and a *C*-terminal core peptide. The leader peptide harbors the sites that recognize the enzymes mediating the post-translational modification of the core peptide. The cleavage of the core peptide from the leader peptide follows the post-translational modification, leading to the formation of an active RiPP that could be transported outside the cell [17].

The genome mining in our study revealed the identification of 60 GCs belonging to nine RiPPs subclasses, namely sactipeptides, lanthipeptides, linear azole-containing peptides (LAPs), head-to-tail cyclized peptides, lasso peptides, proteusin, ComX, lysine-to-tryptophan crosslink, and autoinducing peptides.

##### Sactipeptides

Sactipeptides are post-translationally modified by radical S-adenosylmethionine (SAM) enzymes, resulting in the formation of a thioether bond between sulfur in cysteine residue and the α-carbon of an adjacent amino acid [18].

With 29 GCs found in the selected *Bacillota* genomic sequences, sactipeptides represent the most abundant subclass I. Among the putative sactipeptide GCs identified by BAGEL4, 23 GCs were predicted based on the presence of a modification enzyme, the radical SAM protein (conserved domain PF04055), in the genomes of the following strains: *Geobacillus* sp. 12AMOR1, *Streptococcus thermophilus* APC151, *Paenibacillus durus* DSM 1735 (n = 4), *Paenibacillus donghaensis* KCTC 13049, *Paenibacillus* sp. CAA11 (n = 2), *Desulfotomaculum reducens* MI-1 (n = 2), *Oscillibacter valericigenes* Sjm18-20, *Kyrpidia spormannii* EA-1, *Petrocella atlantisensis* 70B-A (n = 2), and *Anoxybacter fermentans* DY22613 (n = 8) (Appendix A).

The other six sactipeptide GCs were identified based on the similarity of their precursor peptide to thurincin H, subtilosin, and sporulation killing factor (56.41–100%) in the genomes of *Bacillus safensis* KCTC 12796BP, *Bacillus cereus* CC-1, *Bacillus spizizenii* SW83, *Bacillus subtilis* subsp. *subtilis* BS155, *Bacillus anthracis* MCCC 1A01412, and *Staphylococcus delphini* NCTC 12225 (Figure 2 and Appendix A).

Aside from sactipeptide structural genes and the post-translational modification gene *BmbF*, other genes involved in the bacteriocin production machinery were detected, such as ABC transporters, the regulatory system gene (*LanK*), and protease (Figure 2 and Appendix A).

##### Lanthipeptides

Lanthipeptides are composed of lanthionine (Lan) or β-methyllanthionine residues (MeLan) that are biosynthesized through a post-translational modification [3]. The biosynthesis of lanthipeptides includes the dehydration reaction of serine and threonine residues to generate didehydroalanine (Dha) and didehydrobutyrine (Dhb) residues that would be subsequently converted into Lan or MeLan residues via a cyclization reaction [19]. The biosynthetic gene cluster of lanthipeptides contains the gene encoding the core peptide termed *lanA*; modification genes (*lanB*, *lanC*, *lanM*, and *lanKC*) mediating the enzymes involved in the dehydration and cyclization reactions; regulation genes encoding a response regulator (*lanR*) and histidine kinase (*lanK*); and other genes encoding transport (*lanT*), immunity (*lanI (H)* and *lanFEG*), and leader cleavage (*lanP*) [3,20].

The in silico analysis using BAGEL4 revealed that nine putative lanthipeptide GCs were found in the *Bacillota* genomes. Of the lanthipeptide GCs identified, two operons were identified in the genomes of *Geobacillus kaustophilus* HTA426 and *Streptococcus thermophilus* APC151 based on the detection of the genes *LanC* (conserved domain PF05147) and *LanB* (conserved domain PF04738; PF14028), encoding the modification enzyme cyclase and dehydratase in lanthipeptide class I [21]. The operon of *Geobacillus kaustophilus* HTA426 is composed also of five ABC transporters and two regulatory genes, *lanR* and *lanK* (Appendix A).

As a result of the BAGEL4 BLAST hit with known bacteriocin precursor peptides, three lanthipeptide class I operons were found in *Geobacillus kaustophilus* HTA426, *Paenibacillus* sp. CAA11, and *Bacillus spizizenii* SW83 (Figure 3A–C). The first operon contained two structural genes that shared 92.86% identity with geobacillin I and 50% identity with salivaricin D (Figure 3A). The second and third operon contained one structural gene, one that shares 100% identity with streptin and one that shares 100% identity with subtilomycin, respectively. The modification genes *LanC* and *LanB* were located upstream of the precursor peptide in *Paenibacillus* sp. CAA11 and downstream in *Bacillus spizizenii* SW83 (Figure 3B,C). The four other class II lanthipeptide GCs were identified through similarity to cerecidin (76.32%), cytolysin (98.41%; 100%) and lichenicidin A (40.82–84.78%) and through the presence of LanM, modification enzyme of lanthipeptide class II [21] (Figure 3D–F). According to their gene cluster organization, all the class II lanthipeptides harbored the *lanT* gene, which is involved in removing the leader peptide and releasing the mature bacteriocin. The genome screening also predicted ABC transporter, protease, and regulatory genes (*LanR*, *LanK*, and *HisKA*) (Figure 3).

The amino acid sequences of the putative lanthipeptide precursors identified based on homology to known bacteriocins are listed in Appendix A.

##### Linear Azole-Containing Peptides (LAPs)

LAPs are a part of the RiPPs group named thiazole/oxazole-modified microcins (TOMMs), distinguished by their linear structure. The post-translational modification enzymes PznB/C and PznD catalyze, respectively, the cyclodehydration and dehydrogenation of amino acids to produce the thiazole and oxazole heterocycles in LAPs [2,22,23].

The BAGEL4 screening results indicated that the three *Bacillus* species (*B. cereus CC-1*, *B. anthracis* strain MCCC 1A01412, and strain MCCC 1A02161) and *Halobacillus mangrovi* KTB 131 could be potential LAP producers due to the identification of genes involved in the post-translational modification pathway in their genomes, which are the LapBotD enzyme (conserved domain PF02624) and cyclodehydration enzyme (conserved domain PF00881) (Appendix A).

Moreover, according to the sequence identity percentage determined (85.71%), the three *Streptococcus iniae* strains (YSFST01-82, SF1, and QMA0248) were found to harbor an identical core peptide homologous to streptolysin (Appendix A). Their operons contained additional biosynthesis-associated genes, such as modification genes (LapBotD and cyclodehydration enzyme) and ABC transporters (Figure 4).

##### Head-to-Tail Cyclized Peptides

Head-to-tail cyclized peptides are defined as *N*-to-*C*-terminal macrocyclic peptides [2]. The landmark of this RiPP is the stage II sporulation protein M (SpoIIM), which is known also as DUF950 and responsible for both immunity and circularization functions [24].

The in silico analysis showed that five *Bacillota* strains—*Bacillus amyloliquefaciens* SH-B74, *Bacillus* sp. Pc3, *Bacillus velezensis* 9912D, *Geobacillus kaustophilus* HTA426, and *Kyrpidia spormannii* EA-1—have five putative head-to-tail cyclized peptide precursor-coding genes based to their similarities to the core peptides amylocyclicin (100%), circularin (45.07%), and uberolysin (37.84%) (Appendix A). A homologue of DUF950 (conserved domain PF01944) was identified in all the GCs, except the one in *Kyrpidia spormannii* EA-1 (Figure 5).

##### Lasso Peptides

As its name indicates, the lasso peptide is characterized by a lasso-shaped structure, which confers protease resistance to these peptides [7,9]. The lasso-shaped structure is composed of a *C*-terminal linear peptide and a macrolactam ring that is generated by an *N*-terminal amino group coupled with the β- or γ-carboxyl group of Asp or Glu residues [2]. The precursor peptide is encoded by gene *LasA*; post-translationally modified by gene *LasB* and gene *LasC*; and transported by gene *LasD*, which is an ATP-binding cassette transporter [7].

Due to the BAGEL4 mining of the lasso peptide maturation enzyme LasB (conserved domain PF13471) and LasC (conserved domain PF00733), two putative lasso peptide GCs were found in *Bacillus anthracis* MCCC 1A02161 and *Jeotgalibacillus malaysiensis malaysiensis*. Their gene cluster organization also revealed the presence of glycosyltransferase and an ABC transporter (Appendix A).

In addition, two putative lasso peptide precursors were found in *Bacillus cereus* CC-1 and *Paenibacillus crassostreae* LPB0068, displaying, respectively, a 76.19% and 60.87% identity with paeninodin (Appendix A). Upstream of the paeninodin-like encoding gene, the modification gene *LasC* was detected in both gene clusters (Figure 6).

##### Proteusins

Proteusins are linear peptides composed of polytheonamides A and B. These polytheonamides are produced by the epimerization, hydroxylation, and methylation of amino acids. The key genes encoding these extensive post-transitional modifications are *PoyB*, *PoyC*, and *PoyD* [25].

In this study, we identified two proteusin GCs containing homologues to the precursor peptide PoyA (30.30; 35.21%) in the genomes of two *Paenibacillus* species *(P. crassostreae* LPB0068 and *P.* sp. CAA11) (Figure 7 and Appendix A).

##### ComX

ComX is a RiPP involved in the quorum-sensing mechanism of bacilli as a signaling pheromone [23].

The BAGEL4 analysis revealed the presence of two putative ComX GCs, respectively, on the genomes of *Bacillus subtilis* subsp. *subtilis* BS155, sharing a 46.15% identity with ComX1, and in the genome of *Bacillus velezensis* 9912D, sharing a 96.82% identity with ComX4 (Figure 8 and Appendix A).

##### Lysine-to-Tryptophan Crosslink and Autoinducing Peptides

The RiPP harboring a lysine-to-tryptophan crosslink is a cyclized peptide produced by *Streptococcus thermophiles.* This peptide differs from the other macrocyclic peptides according to its structure and biosynthetic pathway [26]. Concerning the autoinducing peptides, they are peptides involved in the quorum-sensing system modulating the auto-production of bacteriocins [27] and the expression of virulence factors [23].

With a 100% identity shared with streptide and autoinducing peptide III, one lysine-to-tryptophan crosslink GC and one autoinducing peptide GC were found in the strains *Streptococcus thermophilus* APC151 and *Staphylococcus aureus* SJTUF_J27, respectively (Appendix A). The lysine-to-tryptophan crosslink operon was encoded for a modification gene and ABC transporter as well (Figure 9A). The autoinducing peptide operon was encoded for a lytic peptide called delta-lysin, alongside the autoinducing peptide III (Figure 9B).

##### Class II

Class II bacteriocins comprise unmodified peptides with a molecular weight less than 10 kDa, and they are commonly produced by lactic acid bacteria and Gram+ bacteria [28,29]. The structural genes are responsible for encoding the precursor peptides that would be cleaved and transported by the ATP-binding cassette (ABC) transporter subsequently. The immunity protein and other accessory proteins are also frequently found on the biosynthetic gene cluster of class II bacteriocins [30].

Class II bacteriocins are classified into four subclasses: subclass IIa contains pediocin-like peptides with a conserved YGNG sequence and a disulfide bridge at their N-terminus; subclass IIb is called two-peptide bacteriocins, referring to their gene cluster harboring two precursor genes encoding two peptides on the same operon; subclass IIc bacteriocins are circular peptides whose precursor gene does not encode an *N*-terminal leader region; and subclass IId regroups the rest of the class II bacteriocins that are unlike the other three subclasses of peptides [2,31].

The genome analysis of 51 *Bacillota* strains resulted in the identification of 16 putative class II bacteriocin GCs. The three GCs identified in the three *Streptococcus iniae* strains (YSFST01-82, SF1, and QMA0248) contained identical precursor peptides that shared 98.28% identity with the thermophilin 13 sequences, a bacteriocin class IIb (Figure 10A). A Blp class IIc bacteriocin operon was identified in the genome of *Streptococcus thermophilus* APC151. The operon encoded for three bacteriocins (blpU, blpK, and blpD), two immunity genes (*EntA Immun*), an ABC transporter, and a *LanT* gene (Figure 10B). The five class IId bacteriocins GCs were predicted in *Lactococcus formosensis* 122061 through the similarity to garvieacin Q (90%), in the three *Streptococcus iniae* strains, and in *Staphylococcus delphini* NCTC 12225 through the identity match to lactococcin 972 (42.10, 44.9%) (Figure 10C,D). Alongside these predicted GCs, a LCI-like encoding gene was found in the genomes of three *Bacillus* spp. (*B.* sp. Pc3, *B. velezensi* 9912D, and *B. amyloliquefaciens* SH-B74) based on the sequence identity (73.91–93.48%) to LCI, an antimicrobial peptide belonging to class II bacteriocin [32] (Figure 10E). Additionally, the strains *Bacillus* sp. Pc3, *Bacillus safensis* KCTC 12796BP, *Paenibacillus durus* DSM 1735, and *Jeotgalibacillus malaysiensis malaysiensis* were found to harbor four GCs containing a precursor peptide that displayed identities ranging between 33.33 and 43.4% to UviB, a class II bacteriocin (Figure 10F).

The amino acid sequences of the putative class II bacteriocin precursors are listed in Appendix A.

##### Class III

The class III bacteriocins are heat-labile peptides larger than 10 kDa and divided into three subgroups: bacteriolysins, non-lytic bacteriocins, and tailocins [2]. Unlike the bacteriolysins, the target of non-lytic bacteriocins is not the cell wall [28]. The tailocins are characterized by their structure, which is similar to the tail of bacteriophages [2]. 

In this study, only one class III bacteriocin GC was found in the genome of *Bacillus spizizenii* SW83, sharing a 40.6% identity with the known class III bacteriocin colicin (Figure 11 and Appendix A)

## 3. Discussion 

Despite the potential of marine bacteria to produce bacteriocins, only a few marine bacteriocins have been identified [12,33]. This could be due to the laborious and time-consuming conventional isolation process. Combining the in silico analysis with heterogonous expression could be a good alternative to produce new bacteriocins. Based on the homology with bacteriocins already identified, the in silico screen helps to identify the potential bacteriocin-producing bacterial strains, the putative bacteriocins gene clusters, and eventually their distribution among bacterial species.

In terms of phylogenetic distribution, we found out that the majority of *Bacillota* strains selected harbors at least one bacteriocin GCs, indicating the widespread nature of putative bacteriocin GCs among the chosen marine *Bacillota* species, and thereby, these marine *Bacillota* strains seem to be prolific producers of bacteriocins. The main interest in marine bacteriocin-producing bacteria is their efficiency for aquaculture applications compared to terrestrial bacteria [34]. According to Santos et al. [35], the marine *Bacillota* phylum is considered one of the potential bioactive metabolites producers. Similarly, our study promotes the bacteriocinogenic potential of the selected marine *Bacillota* species. In terms of phylogeny, with the exception of three strains, all the selected potential strains, both at the intra-species and inter-species classification, were found to encode different numbers and types of bacteriocins GCs. Likewise, previous genome mining studies revealed a diverse distribution of secondary metabolites gene clusters, including bacteriocins, within intra-species strains of *Burkholderia* [36] and of *Lactiplantibacillus plantarum* [37], and within inter-species of *Geobacillus* [3]. In the same context, our study highlights the distinctive trait of every individual strain in terms of harboring bacteriocin gene clusters. It shows the importance of exploring the bacterial strains on an individual basis rather than exploring one reference strain.

The diversity of bacteriocins GCs could be explained by the intraspecific or the interspecific competition for nutriment and space [38,39]. The bacterial competition mediated by the bacteriocin production usually results in the displacement of co-existing sensitive strains [40,41]. In other words, in the same ecosystem, a bacteriocinogenic strain may produce a bacteriocin to inhibit the growth of other co-existing sensitive strains. This can attribute a competitive advantage to the bacteriocin-producing strain, leading it to outcompete other co-existing strains [41], to colonize their microbial community, and to displace the pathogens [31]. For instance, an in vivo study carried on the gastrointestinal tract of a mouse revealed that lantibiotic, a class I bacteriocin, enabled the dominance of its producer, *Blautia producta*, in the colon and thereby the decolonization of the pathogenic strain vancomycin-resistant *Enterococcus faecium* [42]. Another way of acquiring the bacteriocin-encoding genes can be via the horizontal gene transfer (HGT) [41]. The HGT and gene loss might also drive the variation in GCs between related strains [36] and the similarity of GCs between unrelated strains within the same population [43]. Furthermore, the distribution of secondary metabolite GCs, including bacteriocins, can be phylogenetically conserved, as was previously observed in *Bacillus* [32,44,45]. Overall, the prevalence of bacteriocin GCs is strain-specific, and the diversity and distribution of those genes are associated with evolutionary and ecological factors. Nevertheless, the regulation of bacteriocin biosynthesis depends on the presence of competitors, on the fitness and high density of cells, and on the limited nutrient conditions [9,27,46].

The genome screening using BAGEL4 in our study revealed that the 31 potential *Bacillota* genomes were found to encode 1 putative class III bacteriocin, 16 putative class II bacteriocins, and 60 RiPPs-encoding GCs distributed as follows: 29 putative sactipeptide GCs, 9 putative lanthipeptide GCs, 7 putative LAP GCs, 4 putative lasso peptide GCs, 5 putative head-to-tail cyclized peptide GCs, 2 putative ComX GCs, 2 putative proteusin GCs, 1 putative lysine-to-tryptophan crosslink GC, and 1 putative autoinducing peptide GC. Our results indicated the high prevalence of RiPPs gene clusters in the species selected from the marine *Bacillota* phylum and a wide range of RiPPs classes. Among these RiPPs gene clusters, the sactipeptides represent the highest bacteriocin subclass detected. Actually, the production of RiPPs, and especially sactipeptides, is associated with *Bacillota* (*Firmicutes*) [47]. This could be explained by the fact that the majority of the known sactipeptides is produced by *Bacillus* species [48]. Similarly, using another genomic mining approach, Hudson et al. [49] found out that the majority of putative sactipeptides GCs was identified in *Bacillota* (*Firmicutes*), and a few of them were also detected in other phyla, such as *Proteobacteria* and *Bacteroides*. The sactipeptides are characterized by a narrow spectrum of antibacterial activity [50]. The second most abundant RiPPs identified in our study in the marine *Bacillota* genome are the lanthipeptides; their production was also previously associated with *Bacillota* (*Firmicutes*) [19]. The lanthipeptides that have antibacterial activity are called lantibiotics, and they target generally Gram-positive bacteria [50]. As part of the BAGEL4 approaches, predicted bacteriocin GCs are determined based on homology to previously experimentally validated bacteriocins [51]. Experimentally, bacteriocins GCs can be produced through the culturing and isolation of the producing strain, chemical synthesis, and heterologous expression system [52]. As an example of a sactipeptide GC previously reported, two thurincin H homologues were predicted in this study. Thurincin H was firstly produced and purified by culturing the *Bacillota* strain *Bacillus thuringiensis* SF361, leading to the identification of its GC composition [53]. By gene cloning the GC containing the structural genes and other genes involved in transport and immunity in *Bacillus thuringiensis*, thurincin H was successfully expressed and thereby purified in another study [54]. Additionally, using a plasmid-encoded precursor peptide, Wang et al. [55] developed a homologous thurincin H expression system that effectively produces high levels of the bacteriocin from a single structural gene. Heterologous expression, however, was employed in the case of another sactipeptide GC, subtilosin A, for which two homologue peptides were identified in this study. This sactipeptide gene cluster was successfully expressed in *E. coli* by cloning the precursor peptide and the radical SAM enzyme responsible for post-translational modifications [56]. Furthermore, the biosynthetic GC of subtilomycin, a lanthipeptide identified in *Bacillus spizizenii* SW83 in our analysis, was previously detected and purified from *Bacillus subtilis* MMA7, a strain isolated from a marine sponge [57]. As an example of a lasso peptide GC in our study, two paeninodin homologues were predicted in *Bacillus cereus* CC-1 and *Paenibacillus crassostreae* LPB0068. In a study focusing on the lasso peptides that GCs identified in *Bacillota* (*Firmicutes*), constructing a recombinant *E. coli* host carrying a GC from *Paenibacillus dendritiformis* led to the isolation of paeninodin [58]. Using in vitro models, several studies have also confirmed the antibacterial activity of bacteriocins against indicator strains. For instance, thurincin H has previously shown an antibacterial activity against *Bacillus* spp.; *Carnobacterium* spp.; *Geobacillus* spp.; *Enterococcus* spp.; *Staphylococcus* spp.; and *Listeria* spp., such as the human pathogen and *Listeria monocytogens*, related to fish [18,54,59]. Lichenicidin VK21A2, a lanthipeptide identified in our study, was found previously to be effective against *Listeria monocytogenes* and two antibiotic-resistant bacteria, methicillin-resistant *Staphylococcus aureus* and vancomycin-resistant enterococci [60]. Similarly, garvicin Q, a class II bacteriocin predicted by the BAGEL4 analysis, has been reported to have a broad range of antibacterial activity, including pathogenic *Lactococcus garvieae* strains infecting freshwater and marine fish, and thereby causing lactococcosis, an aquaculture hemorrhagic disease [61,62]. In addition to their antimicrobial activity, bacteriocins can act as a signaling peptides by signaling other bacteria or host cells via the quorum sensing system and the immune system, respectively [63]. They can also contribute to the virulence and the dissemination of their pathogenic producer [19]. Therefore, the broad range of putative bacteriocin GCs identified in the marine *Bacillota* genomes in our study may be used in further studies to develop antibacterial metabolites through the expression system in order to regulate bacterial communication and to control the virulence and spread of the pathogenic bacteriocin-producer strain.

In our study, the genome mining of the putative bacteriocin GCs was determined by BAGEL4. BAGEL4 mines the putative bacteriocin operon by detecting the precursor peptide through the BLAST hit of open reading frames to previously characterized precursor peptides in the literature or databases and/or by detecting the context genes associated with the bacteriocin biosynthesis through HMM motifs [5,64], thus helping to identify unknown bacteriocins, especially RiPPs [4,6]. Our results showed that the identified GCs had different organizations. Some GCs contained one or more precursor peptides, along with the biosynthesis genes involved in modification, regulation, immunity, or transport. Other GCs contained only the precursor peptide, while some had the essential biosynthesis context genes but lacked the precursor peptide. The absence of precursor peptide could be explained by the comparatively lower number of certain bacteriocin classes in the BAGEL4 database comparing to others, such as sactipeptide [4], and by the diverse composition of RiPPs genes, leading researchers to overlook identifying uncharacterized precursor peptide using BLAST against databases [6,64]. The alignment and BLAST results to known bacteriocins revealed that some precursor peptides showed high similarity (>90%), while others had either low homology or were not identified. Therefore, this study suggests that the gene cluster that contains a low homologous or unidentified precursor, alongside the context genes, might be a potential uncharacterized bacteriocin that has not been identified yet. Moreover, BAGEL4 follows the general-to-specific approach in genome mining, which starts from the available bacterial strain genomes as input and narrows them down to the specific potential producers [7]. This approach leads to the shortening of the screening time needed for potential bacteriocin producers, and thereby, it saves time and research resources by identifying unknown bacteriocins instead of ultimately discovering previously characterized peptides. However, as the genome mining enables only the prediction of bacteriocin genes and their potential producers, our study should be followed by in vitro or heterologous expression studies to validate and confirm the potential of our findings. For that, the in silico screening of bacteriocins serves as an effective initial step towards identifying new therapeutic metabolites from marine origin and understanding the bacteria–bacteria interactions mediated by bacteriocins.

It is noteworthy that the prevalence of the predicted bacteriocin GCs in this study is reliant on the genome mining tool [17], the abundance of both bacteriocin GCs and their producers in the BAGEL4 database [65], and to the number of marine *Bacillota* species selected from the MarRef database. This study represents an overall bacteriocinogenic potential of 51 *Bacillota* strains, using BAGEL4. Including additional marine genomes databases, as well as bacteriocin and RiPPs genome mining tools, would be advantageous to inclusively explore the marine phylum *Bacillota*.

## 4. Methods

### 4.1. Data Collection

The set of *Bacillota* (*Firmicutes*) bacterial strains selected for this study was obtained from the marine microbial reference genome database MarRef as of 2022 [66]. The RefSeq accession number and the genome sequence files (.fasta) of the 51 selected strains were retrieved from the NCBI (National Center for Biotechnology Information) website. The information about the 51 *Bacillota* strains, including the isolation source, country, and RefSeq assembly accession number, is listed in Appendix A.

### 4.2. Identification of Putative Bacteriocin Gene Clusters

To predict the bacteriocin GCs, the complete genomes of 51 *Bacillota* strains and/or their RefSeq accession numbers were uploaded to the web server BAGEL4, using default parameters. The bacteriocins mining approaches of BAGEL4 are based on the homology of the core peptide of the identified bacteriocins precursor, through the Expected value (E-value) and the identity match, or/and based on the detection of associated context genes involved in bacteriocins biosynthesis through HMM motifs [5,9].

The amino acid sequence of the predicted bacteriocin precursor was aligned with the most similar bacteriocin identified by BAGEL4. The sequence alignment was performed using Clustal Omega [67] in the MUSCLE website [68]. The alignment result was viewed with JalView v2.11.3 [69].

### 4.3. Construction of Phylogenetic Tree of Bacillota Strains

The complete genomes sequences of the 51 *Bacillota* strains, obtained from NCBI, were used to build a phylogenetic tree, using Phylophlan v3.0.67 [70]. Within a supermatrix pipeline of Phylophlan, DIAMOND v0.9.24 [71] was chosen for mapping against the Phylophlan database composed of 400 universal marker genes, followed by MAFFT v7.520 [72] for multiple sequence alignment, TrimAl v1.4 rev15 [73] for the trimming, and RAxML v8 [74] with 100 bootstraps for the tree construction. Thereafter, as an output of Phylophlan pipeline, the resultant tree was visualized by iTOL v6.8.1, an online tool for the display and annotation of phylogenetic tree [75].

The putative bacteriocin GCs identified by BAGEL4 were subsequently merged into the phylogenetic tree, using iTOL v6.8.1.

## 5. Conclusions

In conclusion, the genome mining of the selected *Bacillota* strains using BAGEL4 identified 77 gene clusters (GCs) belonging to different types of bacteriocins, with a high prevalence of RiPPs. Based on the annotation and alignment results, these GCs were found to potentially encode either known bacteriocins or uncharacterized bacteriocins that have not been identified yet. While further in vitro studies will be necessary to experimentally confirm the gene expression and bioactivity of the putative GCs, our findings suggest that the selected marine *Bacillota* can be a potential reservoir for diverse marine bacteriocins that could be an alternative to antibiotics.

## Figures and Tables

**Figure 1 ijms-25-02566-f001:**
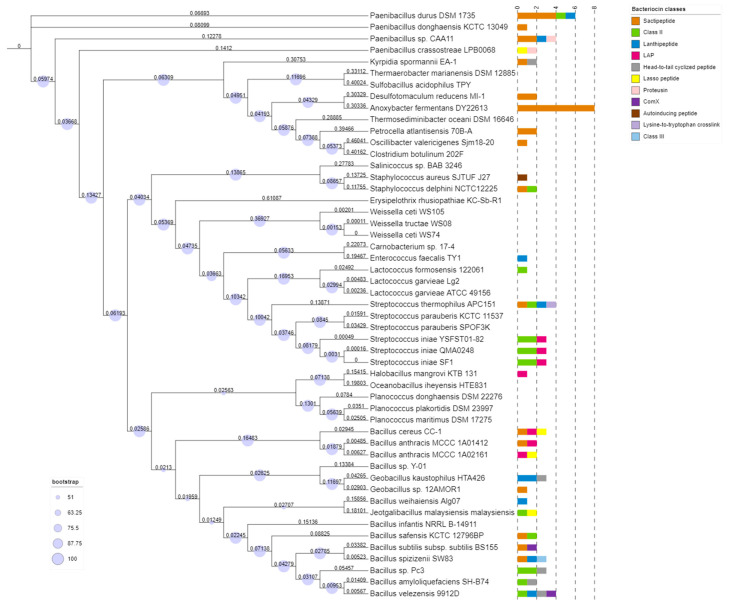
Phylogenetic distribution of 77 putative bacteriocin gene clusters within the 51 complete genomes of marine *Bacillota*. The complete genomes of the corresponding strains were used to create a Maximum Likelihood PhyloPhlAn phylogenetic tree. The numbers on the branches represent the branch lengths. Bar plots on the right represent the number of bacteriocin gene clusters in each strain.

**Figure 2 ijms-25-02566-f002:**
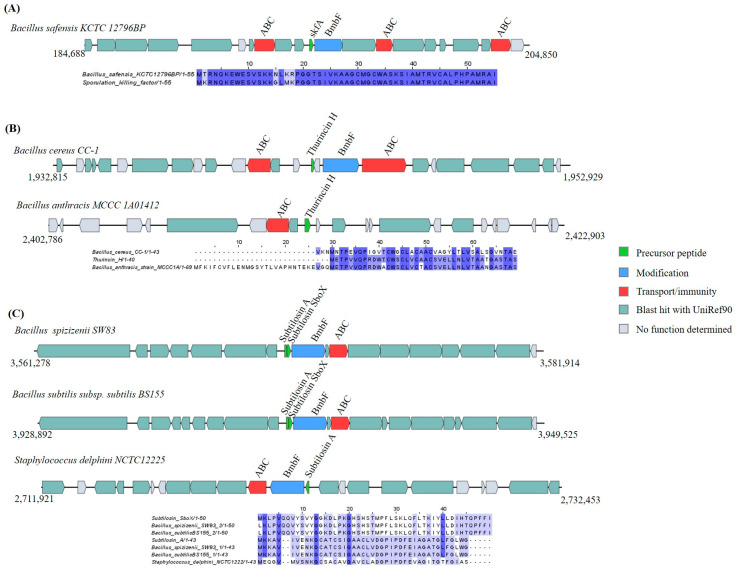
Gene cluster organization of sactipeptides identified in the selected *Bacillota* genomes based on homology to core peptide (examined by BAGEL4) and alignment of the amino acid sequence of the predicted sactipeptide precursor peptide with the sequence of the most similar bacteriocin identified by BAGEL4. (**A**) Sactipeptide gene cluster predicted by homology to sporulation killing factor (skfA) and alignment of predicted precursor peptide to skfA. (**B**) Sactipeptide gene cluster predicted by homology to thurincin H and alignment of predicted precursor peptide to thurincin H. (**C**) Sactipeptide gene cluster predicted by homology to subtilosin and alignment of predicted precursor peptide to subtilosin. The multiple alignment sequence is colored based on the shared identity of the aligned amino acids, with the color shading as the percentage identity increases. No color indicates that the identity is <30%.

**Figure 3 ijms-25-02566-f003:**
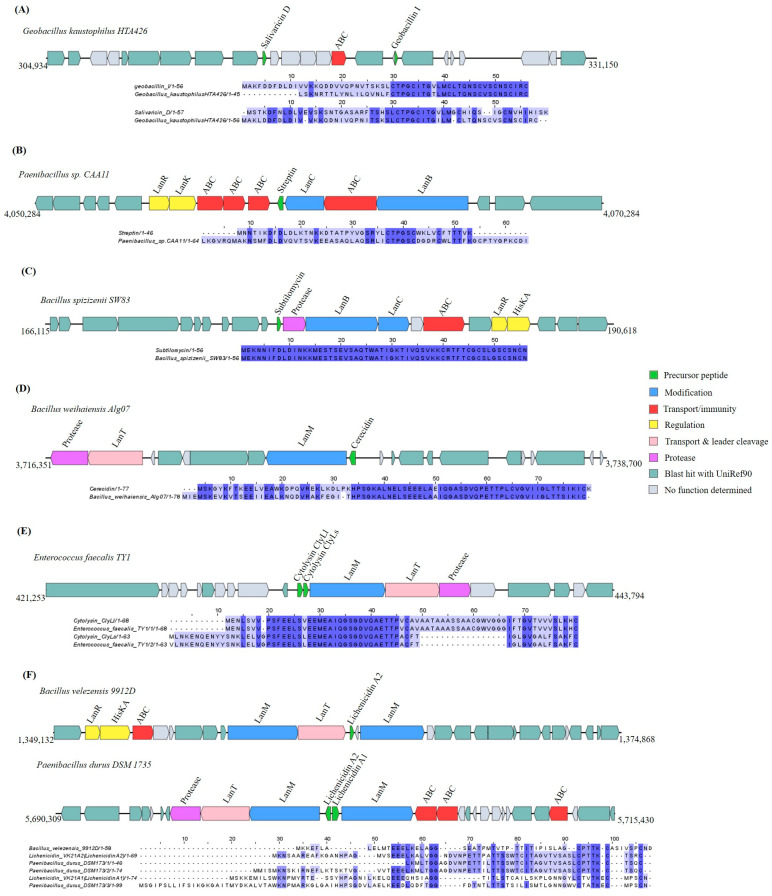
Gene cluster organization of lanthipeptides identified in the selected *Bacillota* genomes based on homology to core peptide (examined by BAGEL4) and alignment of the amino acid sequence of the predicted sactipeptide precursor peptide with the sequence of the most similar bacteriocin identified by BAGEL4. (**A**) Lanthipeptide gene cluster predicted based on homology to geobacillin I and salivaricin D and alignment of the predicted precursor peptide to geobacillin I and salivaricin D. (**B**) Lanthipeptide gene cluster predicted by homology to streptin and alignment of predicted precursor peptide to streptin. (**C**) Lanthipeptide gene cluster predicted by homology to subtilomycin and alignment of predicted precursor peptide to subtilomycin. (**D**) Lanthipeptide gene cluster predicted by homology to cerecidin and alignment of predicted precursor peptide to cerecidin. (**E**) Lanthipeptide gene cluster predicted by homology to cytolysin and alignment of predicted precursor peptide to cytolysin. (**F**) Lanthipeptide gene cluster predicted by homology to lichenicidin and alignment of predicted precursor peptide to lichenicidin. The multiple alignment sequence is colored based on the shared identity of the aligned amino acids, with the color shading as the percentage identity increases. No color indicates that the identity is <30%.

**Figure 4 ijms-25-02566-f004:**
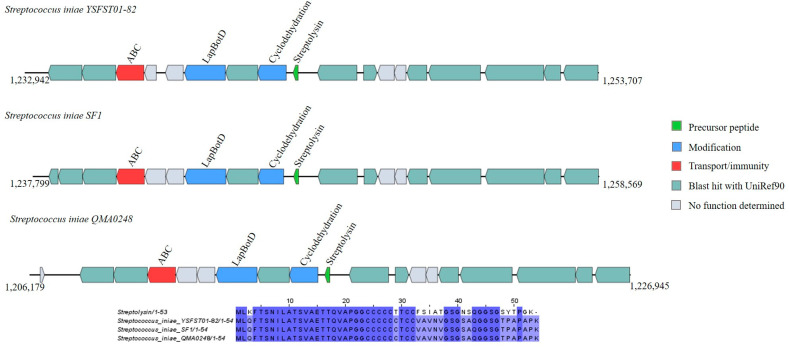
Gene cluster organization of linear azole-containing peptides (LAPs) identified in the selected *Bacillota* genomes based on homology to the core peptide streptolysin (produced by BAGEL4) and alignment of the amino acid sequence of the predicted LAP precursor with the sequence of the most similar bacteriocin identified by BAGEL4. The multiple alignment sequence is colored based on the shared identity of the aligned amino acids, with the color shading as the percentage identity increases. No color indicates that the identity is <30%.

**Figure 5 ijms-25-02566-f005:**
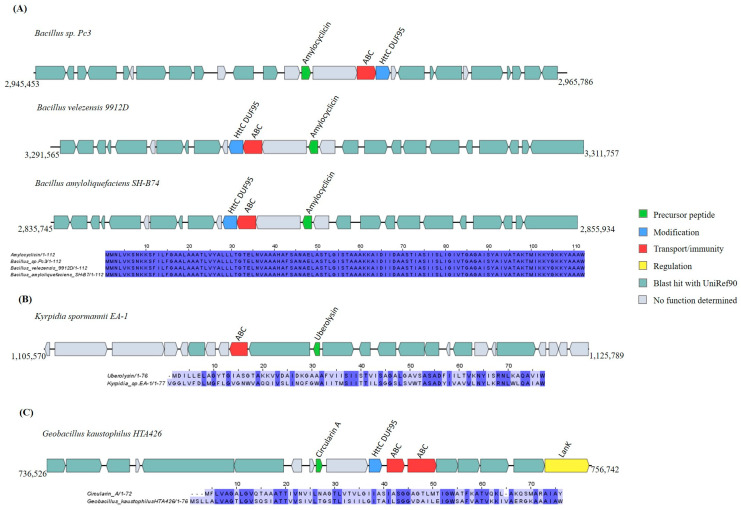
Gene cluster organization of head-to-tail cyclized peptides identified in the selected *Bacillota* genomes based on homology to core peptide (examined by BAGEL4) and alignment of the amino acid sequence of the predicted precursor peptide with the sequence of the most similar bacteriocin identified by BAGEL4. (**A**) Head-to-tail cyclized peptide gene cluster predicted by homology to amylocyclicin and alignment of predicted precursor peptide to amylocyclicin. (**B**) Head-to-tail cyclized peptide gene cluster predicted by homology to uberolysin and alignment of predicted precursor peptide to uberolysin. (**C**) Head-to-tail cyclized gene cluster predicted by homology to circularin A and alignment of predicted precursor peptide to circularin A. The multiple alignment sequence is colored based on the shared identity of the aligned amino acids, with the color shading as the percentage identity increases. No color indicates that the identity is <30%.

**Figure 6 ijms-25-02566-f006:**
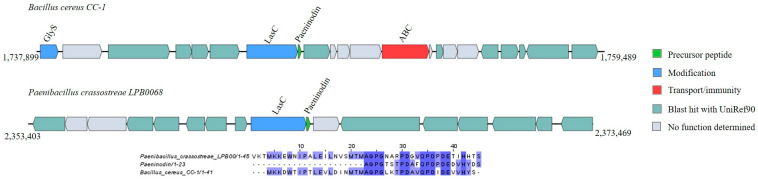
Gene cluster organization of lasso peptides identified in the selected *Bacillota* genomes based on homology to the core peptide paeninodin (examined by BAGEL4) and alignment of the amino acid sequence of the predicted lasso peptide precursor with the sequence of the most similar bacteriocin identified by BAGEL4. The multiple alignment sequence is colored based on the shared identity of the aligned amino acids, with the color shading as the percentage identity increases. No color indicates that the identity is <30%.

**Figure 7 ijms-25-02566-f007:**
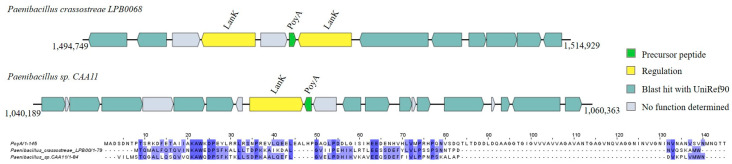
Gene cluster organization of proteusins identified in the selected *Bacillota* genomes based on homology to the core peptide PoyA (examined by BAGEL4) and alignment of the amino acid sequence of the predicted proteusin precursor with the sequence of the most similar bacteriocin identified by BAGEL4. The multiple alignment sequence is colored based on the shared identity of the aligned amino acids, with the color shading as the percentage identity increases. No color indicates that the identity is <30%.

**Figure 8 ijms-25-02566-f008:**
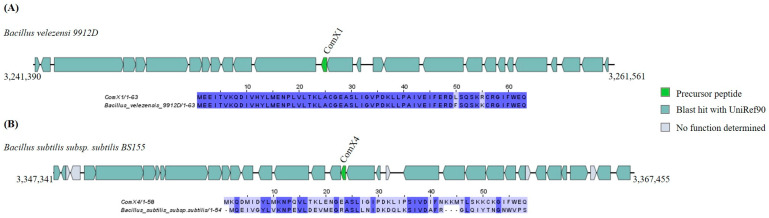
Gene cluster organization of ComX identified in the selected *Bacillota* genomes based on homology to core peptide (examined by BAGEL4) and alignment of the amino acid sequence of the predicted ComX precursor with the sequence of the most similar bacteriocin identified by BAGEL4. (**A**) ComX gene cluster predicted by homology to ComX1 and alignment of predicted precursor peptide to ComX1. (**B**) ComX gene cluster predicted by homology to ComX4 and alignment of predicted precursor peptide to ComX4. The multiple alignment sequence is colored based on the shared identity of the aligned amino acids, with the color shading as the percentage identity increases. No color indicates that the identity is <30%.

**Figure 9 ijms-25-02566-f009:**
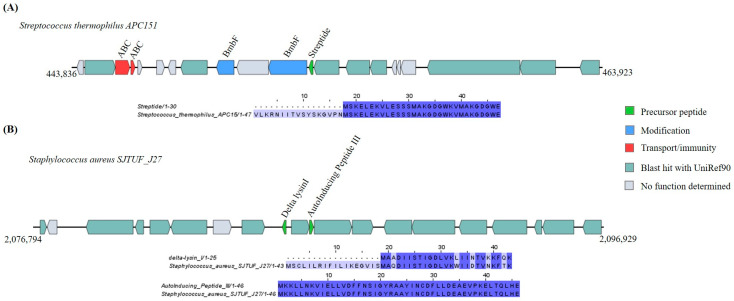
Gene cluster organization of lysine-to-tryptophan crosslink and autoinducing peptides identified in the selected *Bacillota* genomes based on homology to core peptide (examined by BAGEL4) and alignment of the amino acid sequence of their predicted precursor with the sequence of the most similar RiPP identified by BAGEL4. (**A**) Lysine-to-tryptophan crosslink cluster predicted by homology to streptide and alignment of predicted precursor peptide to streptide. (**B**) Autoinducing peptide gene cluster predicted by homology to autoinducing peptide III and alignment of predicted precursor peptide to autoinducing peptide III. The multiple alignment sequence is colored based on the shared identity of the aligned amino acids, with the color shading as the percentage identity increases. No color indicates that the identity is <30%.

**Figure 10 ijms-25-02566-f010:**
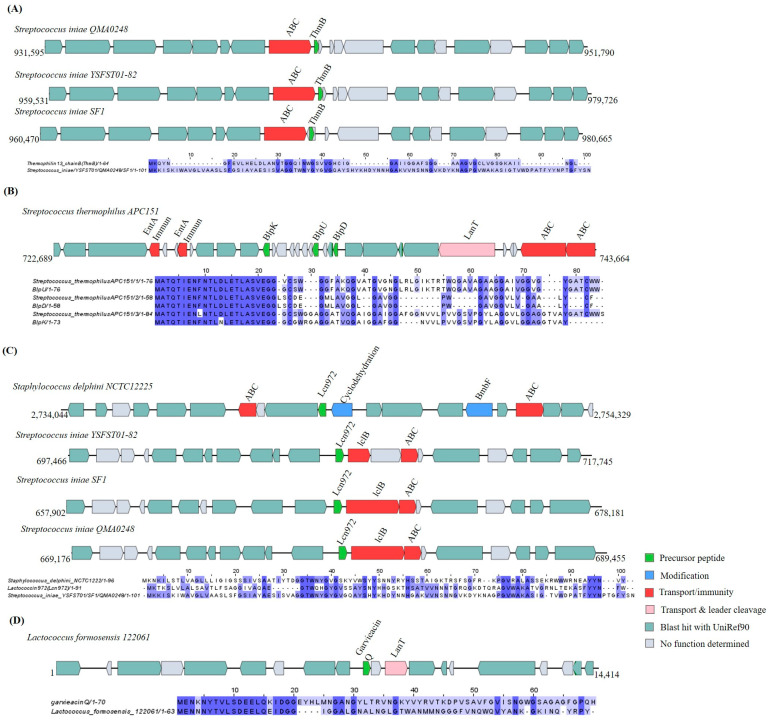
Gene cluster organization of class II bacteriocins identified in the selected *Bacillota* genomes based on homology to core peptide (examined by BAGEL4) and alignment of the amino acid sequence of the predicted precursor peptide with the sequence of the most similar bacteriocin identified by BAGEL4. (**A**) Class II bacteriocin gene clusters predicted by homology to thermophilin 13 (Thmb) and alignment of the predicted precursor peptide to Thmb. (**B**) Class II bacteriocin gene cluster predicted by homology to Blp and alignment of predicted precursor peptide to Blp. (**C**) Class II bacteriocin gene clusters predicted by homology to lactococcin 972 (lcn972) and alignment of predicted precursor peptide to lcn972. (**D**) Class II bacteriocin gene cluster predicted by homology to garvieacin Q and alignment of predicted precursor peptide to garvieacin Q. (**E**) Class II bacteriocin gene clusters predicted by homology to LCI and alignment of predicted precursor peptide to LCI. (**F**) Class II bacteriocin gene clusters predicted by homology to UviB and alignment of predicted precursor peptide to UviB. The multiple alignment sequence is colored based on the shared identity of the aligned amino acids, with the color shading as the percentage identity increases. No color indicates that the identity is <30%.

**Figure 11 ijms-25-02566-f011:**
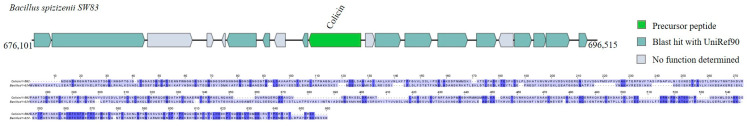
Gene cluster organization of class III bacteriocins identified in the selected *Bacillota* genomes based on homology to the core peptide colicin (examined by BAGEL4) and alignment of the amino acid sequence of the predicted precursor peptide with the sequence of the most similar bacteriocin identified by BAGEL4. The multiple alignment sequence is colored based on the shared identity of the aligned amino acids, with the color shading as the percentage identity increases. No color indicates that the identity is < 30%.

## Data Availability

The original contributions presented in the study are included in the article/Appendix A, further inquiries can be directed to the corresponding author/s.

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
