# Peer review of "In Silico Screening of Bacteriocin Gene Clusters within a Set of Marine Bacillota Genomes"

_ijms, 2024, doi:10.3390/ijms25052566_

Round 1

Reviewer 1 Report

Comments and Suggestions for Authors

Review on “In silico screening of bacteriocin gene clusters within a set of marine Firmicutes genomes” for manuscript ID ijms- 2838454

In this manuscript the authors describe predicted gene clusters of bacteriocins synthesis among marine bacteria (belong to Firmicutes). This new knowledge could help to prepare future experiments and drug research.

In the brief introduction authors describe known classes of ribosomally-synthesized antimicrobial peptides and the approaches to identify them. Unfortunately, some recent studies and experimental evidence of bacteriocins synthesis among marine bacteria have been missed. The following papers could help to improve the Intro:

Uniacke-Lowe S, Collins FWJ, Hill C, Ross RP. Bioactivity Screening and Genomic Analysis Reveals Deep-Sea Fish Microbiome Isolates as Sources of Novel Antimicrobials. Marine Drugs. 2023; 21(8):444. https://doi.org/10.3390/md21080444

Ayikpoe, R.S., Shi, C., Battiste, A.J. et al. A scalable platform to discover antimicrobials of ribosomal origin. Nat Commun 13, 6135 (2022). https://doi.org/10.1038/s41467-022-33890-w

A novel marine bacterium Exiguobacterium marinum a-1 isolated from in situ plastisphere for degradation of additive-free polypropylene // https://doi.org/10.1016/j.envpol.2023.122390

The choice of particular 51 strains is poorly explained (L60), please specify the methodology of your choice (isolation source, taxonomic and geographic stratification). The accession numbers of 6 strains are missing in Table S1. How the genomes were processed if their accession numbers are unknown?

Firmicutes phylum recently was renamed to Bacillota, and please check every species name via database https://lpsn.dsmz.de/advanced_search

My questions about Results and Discussion:

The purpose of the study is not clear. A number of bacteria genomes was analyzed via web-based tool (BAGEL ver. 4), but little new knowledge was obtained. Strain list was limited by MarRef database only.

What the meaning of bar widths at the right part of Figure 1?

Why BACTIBASE Database wasn’t used for mining data about considering strains?

Discussion section lacking comparison of identified gene clusters with experimentally verified ones. Please perform analysis of known experimental studies.

L476: HMM is not a rule-based method, it belongs to machine learning approaches.

Please avoid common words and repetition in the Discussion, see L503-510.

Methods section comments:

Considering strains are close, so the 16S sequence is the poor choice to build a phylogenetic tree. Authors have the whole genome sequences and it’s better to use them for phylogenetic tree construction (f.e. PhyloPhlAn could be used: https://github.com/biobakery/phylophlan)

Any other tools to identify bacteriocin gene clusters weren’t considered, why?

Were the plasmid sequences (where available) have been analyzed?

Some minor corrections to the text (style and spelling):

·        L389, L506: reference needed

·        Multiple font corrections needed, including figure captions

·        L162, L474, L487: “blast” → “BLAST”

·        L436: “genetic screening” → “genome screening”

·        L462: “board range” → “broad range”

Reviewer 2 Report

Comments and Suggestions for Authors

In their manuscript entitled "In silico screening of bacteriocin gene clusters within a set of marine Firmicutes genomes", Teber and Asakawa made an in silico investigation on the potential bacteriocin gene clusters and their potential producers in 51 selected marine Firmicutes using the genome mining tool BAGEL4. About 60% of the selected bacteria were found as potential producers. The authors identified 77 bacteriocin gene clusters, most of them of the class I RiPPs. None of the strains was demonstrate dto produce bacteriocins.

The manuscript is in general well organized and fairly written, the conclusions taken are fair.  There are however some issues that require the attention of the authors:

line 74: define nr.

line 83: "Bacillus ssp." What do the authors mean with this designation. If the authors intend to metion to a sinlge Bacillus species, the correct designation is Bacillus sp.. If th eidea is to mention several Bacillus species, the correct designation is Bacillus spp. There are dozens of designations like this throughout the text. 

Figure 1: The phylogenetic tree was prepared based on the 16S rRNA sequence. Since the authors have used the entire genome sequences of the bacterial strains mentioned, why not to prepare the phylogenetic tree based on the whole genomes sequences? The numbers in the branches indicate what? How many bootstraps were used?

line 89:  bacteriocin

lines 118 to 126: the authors italicized both the species names and the reference like the culture collection reference. Only the species names should appear in italics. Several other examples occur throughout the manuscript and should be corrected.

lines 127 and 129: Genes names should be italicezed, todistinguish between proteins and genes. Other examples occur throughout the text.

line 138: "predicted by homology..."

line 160: "the 2 regulatory..."

line 205: AND?

line 211: a dot is missing

lines 224 to 226: Italicize species names.

line 249: which is an ATP...

lines 324-325 :"encoded on precursor gene, which is cleaved and transported". The sentence makes no sense. as genes are not cleaved only their products, please re-write.

line 460: enterococci is written with a taxonomical significance? If so, should be written as Enterococci spp.. If not no need of italics.

line 493: "strains genomes..."

line 548: "producing a broad rang eof different types of bacteriocins..."

Round 2

Reviewer 1 Report

Comments and Suggestions for Authors

Review on “In silico screening of bacteriocin gene clusters within a set of marine Firmicutes genomes” for manuscript ID ijms- 2838454

I would to thank authors for the efforts to improve the manuscript, but it still requires some corrections.

I strongly suggest to change Firmicutes to Bacillota in the whole manuscript, while the recent papers use updated taxonomy, including these reviews https://doi.org/10.3390/ijms242317022 and bacteriocins studies https://doi.org/10.3390/ijms242115644, https://doi.org/10.3390/md21110569  

My questions about Results and Discussion:

Please extend the Discussion section with experimentally verified gene clusters of bacteriocins synthesis of Bacillota (Firmicutes) representatives. This way authors could support the predictive role of genome mining tools. A recent review could help https://doi.org/10.3390/md21110569

Please consider to rewrite Conclusion to better summarize the key findings of the study and avoid common words.

Methods section comments:

Why the genome accessions weren’t defined in Table S1?

Geobacillus sp. 12AMOR1 was published https://doi.org/10.1186/s40793-016-0137-y, but assembly was suppressed due to chimeric nature of the sequence https://www.ncbi.nlm.nih.gov/datasets/genome/GCF_001028085.1/

Please provide all assembly accessions (GenBank, if RefSeq isn’t available).

Some minor corrections to the text (style and spelling):

·        L452: missing space

·        L502: “using blasting” → “using BLAST against databases”

Reviewer 2 Report

Comments and Suggestions for Authors

The critiisms raised were mostly answered in the revised version provided by authors. There is only one remaining issue that needs to be solved. In fact, branch values cannot correspond to branch length. Please correct this issue.
